# EEG microstate transition cost correlates with task demands

**Giacomo Barzon**[1,2]*, **Ettore Ambrosini**[1,3], **Antonino Vallesi**[1,3], **Samir Suweis**[1,4]

**1** Padova Neuroscience Center, University of Padova, Padova, Italy, **2** Fondazione Bruno Kessler, Povo, Italy, **3** Department of Neuroscience, University of Padova, Padova, Italy, **4** Department of Physics and Astronomy "Galileo Galilei", University of Padova, Padova, Italy

* giacomo.barzon.1@phd.unipd.it

## Abstract

The ability to solve complex tasks relies on the adaptive changes occurring in the spatio-temporal organization of brain activity under different conditions. Altered flexibility in these dynamics can lead to impaired cognitive performance, manifesting for instance as difficulties in attention regulation, distraction inhibition, and behavioral adaptation. Such impairments result in decreased efficiency and increased effort in accomplishing goal-directed tasks. Therefore, developing quantitative measures that can directly assess the effort involved in these transitions using neural data is of paramount importance. In this study, we propose a framework to associate cognitive effort during the performance of tasks with electroencephalography (EEG) activation patterns. The methodology relies on the identification of discrete dynamical states (EEG microstates) and optimal transport theory. To validate the effectiveness of this framework, we apply it to a dataset collected during a spatial version of the Stroop task, a cognitive test in which participants respond to one aspect of a stimulus while ignoring another, often conflicting, aspect. The Stroop task is a cognitive test where participants must respond to one aspect of a stimulus while ignoring another, often conflicting, aspect. Our findings reveal an increased cost linked to cognitive effort, thus confirming the framework's effectiveness in capturing and quantifying cognitive transitions. By utilizing a fully data-driven method, this research opens up fresh perspectives for physiologically describing cognitive effort within the brain.

**Data Availability Statement:** The preprocessed data and the code that support the findings of this study are available at https://github.com/gbarzon/brain_control_cost and were deposited on Zenodo (https://doi.org/10.5281/zenodo.13709700).

## Author summary

In our daily lives, our brains manage various tasks with different mental demands. Yet, quantifying how much mental effort each task demands is not always straightforward. To tackle this challenge, we developed a way to measure how much cognitive effort our brains use during tasks directly from electroencephalography (EEG) data, which is one of the most used tools to non-invasively measure brain activity. Our approach involved the identification of distinct patterns of synchronized neural activity across the brain, named EEG microstates. By employing optimal transport theory, we established a framework to quantify the cost associated with cognitive transitions based on modifications in EEG microstates. This allowed us to link changes in brain activity patterns to the cognitive effort

**Funding:** Work by A.V and S.S. is supported by #NEXTGENERATIONEU (NGEU) and funded by the Ministry of University and Research (MUR), National Recovery and Resilience Plan (NRRP), project MNESYS (PE0000006) – A Multiscale integrated approach to the study of the nervous system in health and disease (DN. 1553 11.10.2022). The funders had no role in study design, data collection and analysis, decision to publish, or preparation of the manuscript.

**Competing interests:** The authors have declared that no competing interests exist.

required for task performance. To validate our framework, we applied it to EEG data collected during a commonly employed cognitive task known as the Stroop task. This task is recognized for challenging us with varying levels of cognitive demand. Our analysis revealed that as the task became more demanding, there were discernible shifts in the EEG microstates. Importantly, these shifts in neural activity patterns corresponded to higher costs associated with cognitive transitions. Our approach offers a promising methodology to assess cognitive effort using neural data, contributing to our comprehension of how the brain manages and adapts to varying cognitive challenges.

## Introduction

The complex activity patterns that support perception, cognition, and behavior in the healthy brain arise from the interactions of neuronal populations across various spatial and temporal scales [1]. At the macroscale, brain activity is characterized by spatially distributed groups of regions that exhibit temporally correlated activity and co-activate during behavioral tasks, thus acting as functional networks [2]. Recently, it has been shown that such functional networks may reflect the long-time average of rapidly switching metastable patterns (also called "metastable substates" or "dynamical states"), which are consistently observed with different imaging methods [3–7]. In the M/EEG literature, these patterns are termed "microstates" and are highly reproducible across studies and clustering techniques [8–10].

As our environment is constantly evolving, with new stimuli and challenges emerging regularly, our brain must remain flexible and adaptable to respond effectively to these changes. A crucial component that drives such reconfiguration is "executive functioning" or "cognitive control" [11–13]. This construct refers to the set of processes and mechanisms that enable goal-directed behavior in the face of changing circumstances [11,14–16]. When confronted with challenging situations, cognitive control allows the brain to regulate attention, inhibit irrelevant information, and shift cognitive resources to prioritize relevant tasks or goals [17].

Concurrently, it has been shown that the dynamical properties of the metastable substates during different active conditions are modulated compared with the resting state. Such adaptation has been demonstrated in a large variety of conditions such as cognitive loads [18], sleep-awake cycle [19], habituation of cognitive tasks [20], and is reflected in the overall reconfiguration of functional connectivity [21–23]. Importantly, alterations in the dynamic of brain states were found in psychiatric [8,24] and neurological disorders [25] and during normal aging [26]. Therefore, developing quantitative measures for quantifying the cost of such reconfiguration in the brain is crucial for explaining the impairments and guiding the possible effects of therapeutic interventions [27].

In recent years, much attention has been captured by the network controllability framework for measuring the brain transition cost [28,29]. Control theory based tools offer a mechanistic explanation for how the brain moves between cognitive states drawn from its network organization. In addition, control theory provides a quantitative way of computing the control cost as the amount of energy needed to steer a system along a desired trajectory. Despite its potential and broad spectrum of applications, it has some strong limitations [30,31]. For instance, it relies on the assumption of linearity in the dynamics. However, linear models fail to capture non-linear [32] and higher-order [33] phenomena ubiquitously encountered in brain dynamics. Moreover, stochasticity is not considered, but it is essential for accurately describing many aspects of brain function [34].

A promising approach for circumventing these limitations in quantifying the cost of control consists of reframing the task into a Schrödinger bridge problem [35,36]. More specifically, given an initial and a target probability distribution, representing, for instance, the distribution of metastable substates during resting and task conditions, the Schrödinger bridge problem asks for the most likely path or "bridge" that connects the two probability distributions given the spontaneous (resting) stochastic dynamics of the system. The transition cost is then estimated as the Kullback-Leibler divergence, which measures distances in the probability distribution space, between the baseline trajectory and the bridge. Intuitively, it measures the cost of "transporting" one distribution into another by a stochastic process that satisfies some given constraints. Indeed, the Schrödinger bridge problem has been proven to be formally equivalent to an (entropy-regularized) optimal transport problem [37,38].

Recently, such an approach was applied to an fMRI dataset of participants performing several cognitive tasks [39]. The authors show that the transition cost from the resting condition to the various tasks varies significantly, thus proposing this approach might be suitable for describing neurophysiological data. However, the tasks were qualitatively different and difficult to compare, thus there were no strong prior expectations of the task difficulty and the expected cognitive demand. Additionally, the reliability of individual differences in task-based fMRI activity is known to be quite poor [40], especially in the absence of long time series, as typically occurs in fMRI data, thus hindering the possibility of a subject-level analysis. Hence, in [39] time series data from individual subjects were combined to create a unified meta-subject dataset. Therefore, the analysis was exploratory and the reliability of such a metric remains limited. Moreover, in many contexts of cognitive interests, fMRI is not a suitable tool to measure neural correlates of behavior. For instance, one of its primary constraints is its inherent limitation to cognitive tasks that do not involve significant physical movement. This is a notable drawback, as many cognitive processes and behaviors inherently entail motor actions.

In this work, we bridge this gap by generalizing the above method to electroencephalography (EEG) signals, which moreover measure neural activity more directly than fMRI. Specifically, we analyze an EEG dataset on participants performing a spatial Stroop task. The Stroop task is a standard experimental paradigm in cognitive psychology that investigates different aspects of cognitive control and executive functions, including selective attention, response inhibition, and interference resolution, by assessing the interference effect from conflicting stimulus features [41]. In its spatial variant [42–44], participants are typically presented with arrows pointing in different directions (e.g., top left or bottom right) and are asked to indicate the direction of the arrow through a spatially compatible button press. However, the pointing direction may conflict with its spatial location. For example, an arrow pointing to the top left corner might appear on the bottom right side of the screen. Typically, participants are slower and less accurate in incongruent conditions (i.e., when the spatial location of the arrow conflicts with the direction it is pointing to) than in congruent conditions (i.e., when spatial location and pointing direction coincide). This interference effect, referred to as the "Stroop effect", is believed to reflect the difficulty in suppressing the automatic processing of the spatial location of the stimulus in favor of the task-relevant information (the arrow direction), with the consequent activation of a wrong response code that then needs to be suppressed. Commonly, it is computed as the difference in the response time (RT) between incongruent and congruent trials. Cognitive control demands can be further manipulated by varying the proportion of congruency (PC), namely the proportion of congruent trials in a given task block [45–47]. Indeed, in high-PC blocks, conflict is less likely, and cognitive control demands are lower, whereas, in low-PC blocks, trials are mostly incongruent, and cognitive control is more required [45,48]. Therefore, due to our well-defined quantitative prior expectation of cognitive

demands, this dataset is ideally suited for assessing the effectiveness of the proposed framework to estimate brain transition costs.

Here, we first characterize the dynamics with a microstate analysis, which reveals that different conditions modulate the distribution of microstates. Next, we calculate the transition cost for each participant from the resting state to the various conditions. We observe a higher cost for incongruent stimuli. Importantly, this cost is significantly influenced by the level of cognitive control. Moreover, we find a correlation between variations in the cost and RTs, showing that a reduced cost is associated with improved task performance. Overall, these results highlight the value of characterizing brain dynamics and transition costs in understanding cognitive processes and offer insights into the relationship between neural activity patterns, cognitive effort, and behavioral performance.

## Results

### Framework for computing the control cost

In this work, we analyzed an EEG dataset recently collected (see "Dataset" section; Fig 1A). This dataset encompasses EEG recordings collected from a cohort of 44 participants during both a 5-min eyes-open resting-state session and a task-oriented sessions. Specifically, the task involved a spatial Stroop task designed with blocks featuring three distinct PC values (25%,

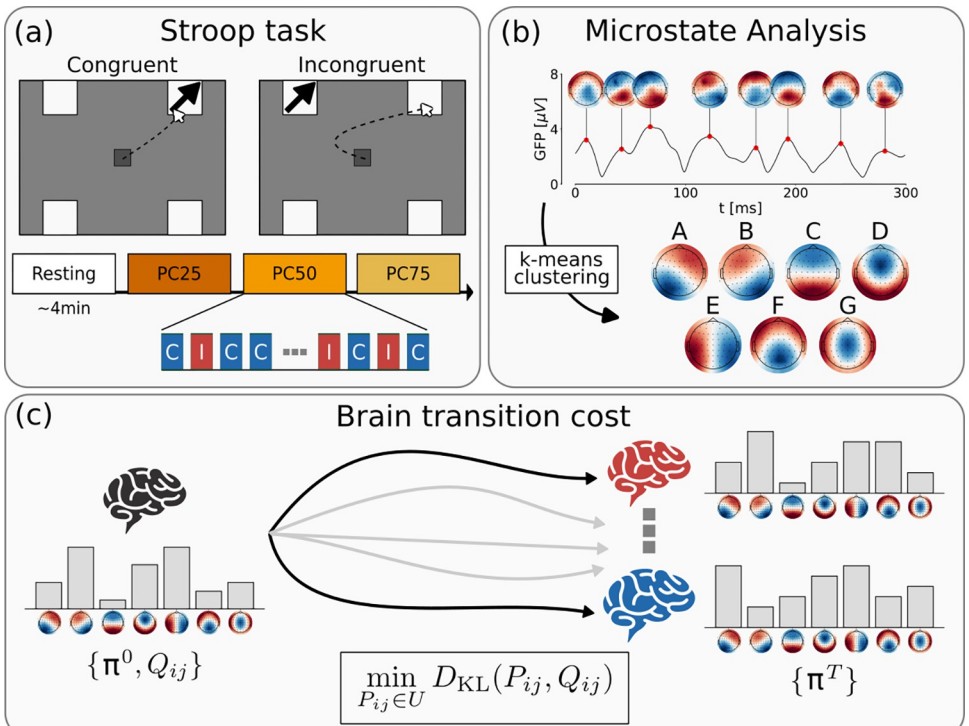

**Fig 1. Summary of the framework for the computation of the brain transition cost from EEG data.** *(a)* The EEG activity of 44 participants was acquired at rest and while performing a spatial Stroop task. The participants were presented either with a congruent (C) or incongruent (I) stimulus. The proportion of congruency (PC) was modulated within three blocks (PC25: 25% C, 75% I; PC50: 50% C, 50% I; PC75: 75% C, 25% I). *(b)* EEG activity was characterized by employing a microstate analysis. The modified k-means clustering found seven most representative topologies, which we named from A to G. *(c)* Schrödinger bridge framework for computing brain transition cost. Given the microstate probability distribution at rest ($\pi^0$) and while performing a task ($\pi^T$), the cost is computed as the Kullback-Leibler divergence between the spontaneous (resting) dynamics, described by joint probability for two consecutive steps ($Q_{ij}$), and the Schrödinger bridge, i.e., the most probable path that links the resting and task distribution, subject to the given constraints.

50%, and 75%) to systematically manipulate different levels of cognitive control engagement (high, medium, and low, respectively). Consequently, this setup provided multiple pre-established levels of cognitive demand expectation.

EEG activity was characterized utilizing a microstate analysis (see "EEG microstate-based analysis" section; Fig 1B). After identifying the most reliable templates for group maps, we proceeded to assess their distributions (also called "coverage" in the microstate literature) and transitions in each participant during both the resting and task conditions.

Utilizing their dynamics, we derived an estimation of the control cost employing the Schrödinger bridge framework (see "Brain transition cost" section, Fig 1C). In essence, this cost was calculated as the disparity between the spontaneous microstate dynamics during the resting phase and the bridge, which corresponds to the most likely pathway linking the distributions of microstates observed during resting and task-oriented conditions.

## Microstate reconfiguration during task

The group-level clustering revealed seven optimal microstate classes, which explained almost 80% of the variance of the dataset (S1 Fig). These group maps resembled the usual microstates template ubiquitously found in the literature [49], and we labeled them accordingly (from A to G).

Before entering into applying the control cost framework, we needed to verify whether microstate distributions during the task were modulated with respect to the baseline (Fig 2). To achieve this, we first compared the microstates distribution at rest to the chance probability reflecting a uniform distribution across the seven microstates (i.e., $1/7 = .1429$; see "Statistical Analysis" section). We found that microstate D was significantly more expressed at rest ($M = 0.158$, $SE = 0.003$, $t(43) = 4.53$, $p < .0001$, $d = 0.68$), while microstate E was significantly

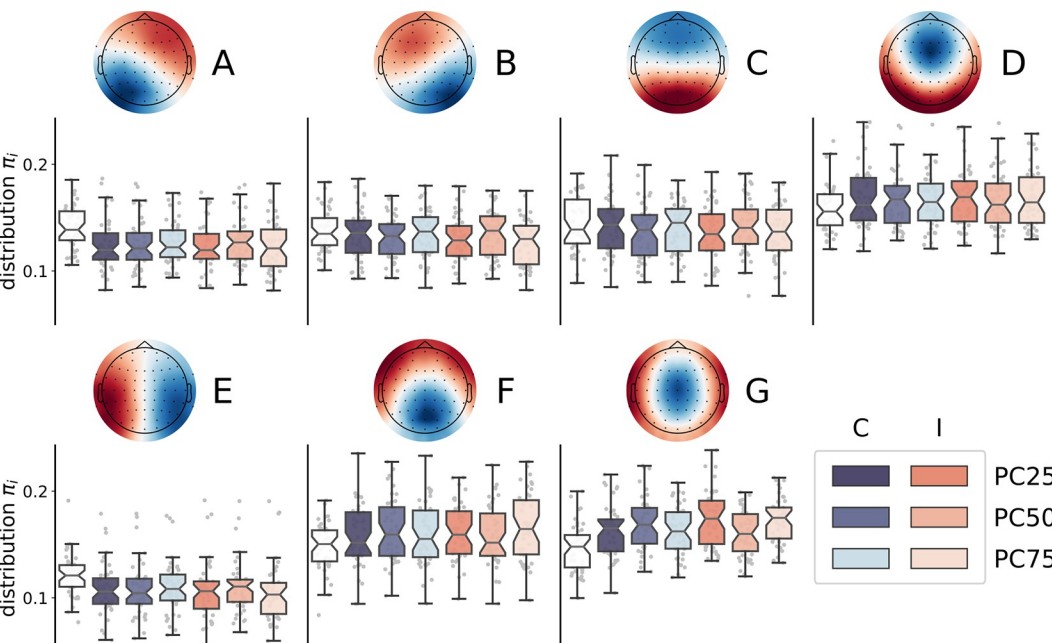

**Fig 2. Microstate distributions distinguish tasks from resting.** The boxplots show the microstate distributions at rest (white boxplot) and during the experimental conditions. The saturation of the blue (orange) scale represents the PC level (and, thus, the level of control demands).

less expressed at rest ($M$ = 0.121, $SE$ = 0.003, $t(43)$ = -7.54, $p$ < .0001, $d$ = -1.14). Next, we performed a linear mixed effects model for each microstate, incorporating congruency, PC level, and their interaction as predictors (See "Statistical Analysis" section). As a dependent variable, we computed the change in the probability distributions across different conditions compared to the resting state. Consequently, the intercept denoted an overall modulation from the resting state to task execution. The results of these analyses are reported in S1 Table. We found that microstates A and E were significantly suppressed during the execution of the task (intercept effect: $b$ = -0.018 and -0.012, $SE$ = 0.004 and 0.003, respectively). Moreover, their expression was significantly modulated by congruency, with a stronger suppression for incongruent compared to congruent trials ($b$ = 0.003 and 0.003, $SE$ = 0.001 and 0.001, respectively), as well as by the congruency by PC interaction ($b$ = 0.005 and 0.004, $SE$ = 0.002 and 0.002, respectively). These interaction effects were explained by the fact that the expression of both microstates was modulated by the PC level in opposite ways, being it less suppressed in congruent trials and more suppressed in incongruent trials as the PC level increased, resulting in an increase of the Stroop effect (i.e., the difference between incongruent and congruent trials) at higher PC levels, that is, when the cognitive control demands were lower. A similar pattern was observed for microstate B, with significant effects of congruency and the congruency by PC interaction ($b$ = 0.005 and 0.005, $SE$ = 0.001 and 0.002, respectively), despite the intercept effect did not survive the correction for multiple tests ($b$ = -0.006, $SE$ = 0.003). Microstate C also showed a significant effect of congruency, with a more suppressed expression in incongruent compared to congruent trials ($b$ = -0.005, $SE$ = 0.001), and was generally suppressed compared to rest ($b$ = 0.013, $SE$ = 0.002). By contrast, microstate G showed the opposite pattern of results compared to microstates A and E, being its expression significantly enhanced during task execution ($b$ = 0.021, $SE$ = 0.002) and significantly modulated by both congruency, with a stronger enhancement for incongruent compared to congruent trials ($b$ = -0.010, $SE$ = 0.001), and the congruency by PC interaction ($b$ = -0.006, $SE$ = 0.003), with positive and negative PC effects for incongruent and congruent trials, respectively. Finally, microstate F was significantly enhanced in incongruent compared to congruent trials ($b$ = 0.005, $SE$ = 0.001), microstate D was significantly more expressed overall compared to rest ($b$ = 0.010, $SE$ = 0.003). The remaining effects were not significant after correction for multiple tests (see S1 Table).

## Transportation cost matrix

A key quantity, that we inferred from the EEG time series and their microstates, was the transportation cost matrix (Fig 3A). Such a matrix, in an optimal transport problem, provides information about the costs associated with transporting goods or resources from one location to another. In our framework, it defines the cost associated with increasing or decreasing the probability of one microstate from the source to the target distribution, and it has a clear intuition: the transportation cost is minimized along the more favorable transitions (i.e., more probable) during rest.

For each participant, the transportation cost was obtained from the joint probability distribution of co-occurrence for two consecutive steps during resting. As shown in Fig 3B, we found that such distribution is asymmetric, indicating a preference or bias in transitioning from one state to another with respect to the opposite direction. Moreover, the self-transition probabilities are quite large, indicating that the system tends to persist in its current state over time, thereby confirming the metastable nature of the microstates.

## Transition cost reflects task demand

Subsequently, we investigated for each participant the costs associated with transitioning from a resting state to different conditions within the Stroop task. To quantify these costs, we

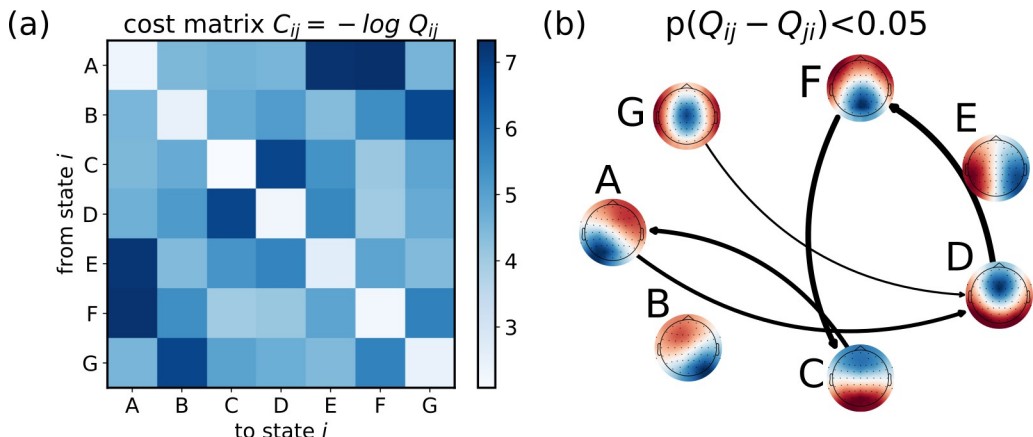

**Fig 3. Estimating the transportation cost matrix from microstate joint probability of consecutive timesteps at rest.**
(*a*) Transportation cost matrix, averaged over the 44 participants, representing the cost for the brain to transition from state *i* to state *j*. (*b*) Network describing the transitions among microstates during resting. We show only the significant asymmetric transitions (t-test, p<0.05).

utilized the Schrödinger bridge framework and calculated the associated Kullback-Leibler divergence ([Fig 4A]). We again performed a linear mixed effect model with the interaction between Congruency and PC on the computed transition costs (see "Statistical Analysis" section). This analysis revealed the significant effects of congruency, with lower costs for congruent than incongruent trials ($b$ = -0.007, $SE$ = 0.001, $t$ = -5.04, $p < .0001$, $d$ = -0.76), and the congruency by PC interaction ($b$ = -0.009, $SE$ = 0.002, $t$ = -3.74, $p = .0001$, $d$ = -0.42), with positive and negative PC effects for incongruent and congruent trials, respectively.

To examine the potential relationship between transition costs and task performance of each participant, we performed a linear mixed effects model on the computed costs including response times as predictors after having calculated the difference in costs and response times between incongruent and congruent conditions (i.e., the Stroop effects), considering each level of control (see "Statistical Analysis" section; see also Figs [4B] and [S2]). Our results revealed that higher Stroop effects in transition costs were associated with higher Stroop effects in response times and potentially indicative of performance ($b$ = 0.054, $SE$ = 0.015, $t$ = 3.69, $p = .0003$, $d$ = 0.45).

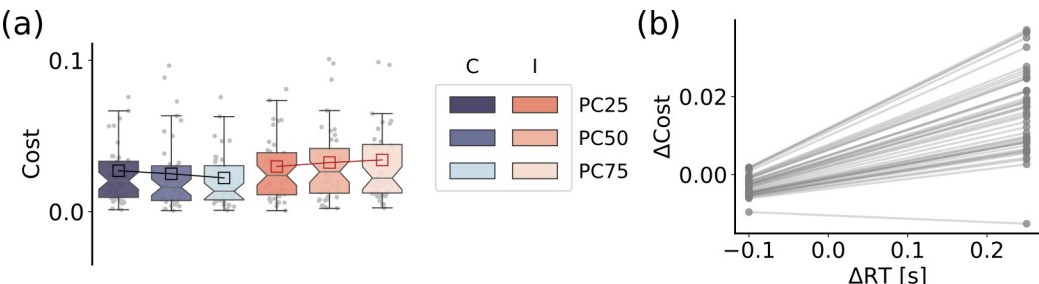

**Fig 4. Brain transition cost correlates with task demand and performance.** (*a*) The boxplots show the distribution of transition costs as a function of both the stimulus congruency, that is congruent (C) vs. incongruent (I) and the PC level (PC25, PC50, PC75;), as indicated by their significant interaction in the linear mixed effects analysis. (*b*) The plot shows the participants' Stroop effects in transition costs (i.e., the difference in transition costs between incongruent and congruent trials: Δ Cost) as a function of their Stroop effects in response times (Δ RT), as indicated by the random effects revealed by the linear mixed effects model (see main text).

We performed the same analyses on the Kullback-Leibler divergence between the microstates distributions during the task and resting conditions instead of transition costs (see S3 Fig). These analyses failed to find the significant effect of congruency ($t$ = -1.74, $p$ = .0833, $d$ = -0.26) and the congruency by PC interaction ($t$ = -1.33, $p$ = .1841, $d$ = -0.20) on Kullback-Leibler divergence, and the Stroop effects in Kullback-Leibler divergence values were not significantly associated with participants' performance, as assessed by their Stroop effects in response times ($t$ = 0.19, $p$ = .8467, $d$ = 0.03).

## Discussion

In this study we have employed a stochastic control framework to measure the brain transition cost in an existing EEG dataset. Through our investigation, we have confirmed a correlation between cost and cognitive demand observed during a spatial Stroop task. To our current knowledge, this is the first application of such a framework to EEG data, thus providing a computational pipeline to quantify cognitive demand in EEG experiments.

To estimate brain transition costs from the Schrödinger bridge framework, we used a probabilistic approach that resorts to a reduction of dimensionality. A growing body of literature suggests that brain activity, across different scales, exhibits organization within a low-dimensional manifold. The trajectories of neural activity can thus be described as discrete transitions between a few metastable attractors, which capture a significant portion of the overall activity variance. In particular, the analysis of EEG activity is increasingly conducted using the microstates approach. This method reduces the recorded electrical signal into non-overlapping and distinct topographies [8,9]. Although individual topographies have been associated with partial activations or deactivations of canonical resting-state networks [49,50] and specific spectral bands [51,52], the functional and cognitive role of the microstates has not yet been fully established [10].

Specifically within our dataset, we observed distinct distributions of microstates across different conditions. Notably, certain microstates differentiated between tasks and resting states, while others were specific to incongruent stimuli and modulated by the level of cognitive demand. Specifically, microstate C was suppressed during the tasks, possibly representing posterior dominant resting state (alpha) rhythms. Also microstates A and E were suppressed during the task, with larger suppression for a larger level of cognitive demand. As a tentative speculation, they might be related to alpha suppression or the deactivation of the default mode network. However, this interpretation is in contrast with existing literature, since microstate A is commonly attributed to resting condition [8,50], but it should be noted that we used an eyes-open resting state, which likely reduced the power in the alpha band. Instead, microstates D, F, and G were more prominent during tasks. Microstate F was more prevalent during incongruent stimuli compared to congruent stimuli, while microstate G was more present in blocks with higher expected levels of control. Consequently, they may be linked to specific brain regions involved in inhibitory control and conflict resolution [53,54].

Moreover, a global modulation can be assessed by computing the Kullback-Leibler (KL) divergence between the task and resting conditions [18]. We observed that the KL divergence tends to be larger for more demanding task conditions, which is consistent with the idea that cognitive load influences the divergence between microstate distributions. However, despite this trend, our analyses failed to find significant effects. This suggests that while KL divergence is a useful measure for capturing global shifts between task and resting states, it may lack the sensitivity required to detect more nuanced effects related to cognitive cost when compared to the Schrödinger bridge method. The latter, by accounting for the transition paths between distributions, appears to capture additional information about cognitive load that is not reflected in KL divergence alone.

This modulation of certain microstates could be associated with the dynamic reorganization of specific functional networks, as previously observed [55,56]. Confirmation of these hypotheses and further investigation into the microstates can be achieved through source localization, which will be explored in future works. Overall, a higher cost may be related to a larger network reconfiguration. Indeed, a larger cognitive demand induces a more global alteration in brain activity, which is needed to make functional networks transiently adopt a more efficient but less economical configuration [21]. However, the specific mechanisms governing these shifts between states of the brain remain unclear [57,58]. Moreover, whether such cognitive cost may represent an increase in metabolic consumption is still to be investigated [59]. It is important to mention that in a stochastic linear setting, the Schrödinger bridge control cost is formally equivalent to the "classical" control cost (i.e., the expectation of the time integral of squared control signal) [37,39,60], which has a clear physical interpretation.

Our approach integrates into the current literature on the brain's neural control [28–31,34,57,58,61–63]. The core foundation of all these models involves a metric that quantifies the amount of effort required for a dynamical system to traverse its state space across diverse conditions. The existing methodologies typically rely on the full knowledge of the underlying structural connectome and an explicit representation of the dynamics. Under the assumption of linear dynamics, it is possible to estimate this metric efficiently by utilizing an explicit analytical formula [28,64]. However, these approaches overlook the intricate nonlinear characteristics of brain dynamics, and may not be computationally feasible for large networks [30]. On the contrary, to extend this framework to biophysically detailed dynamical models, extensive numerical simulations become a necessary recourse [19]. Instead, our approach offers the advantage of estimating the reconfiguration cost directly from neurophysiological recordings. Additionally, its flexibility allows for versatile application across various imaging techniques [39]. However, its applicability to EEG data holds particular importance due to its widespread usability, cost-effectiveness compared to techniques such as fMRI or MEG, and non-invasiveness compared to intracranial recordings.

We also found that the probability of transitioning from one microstate to another is asymmetric. Such asymmetric transitions indicate potential fluxes and net flows within the system, which in turn can contribute to the overall production of entropy at a macroscopic level [65]. Therefore, our results hint at a macroscopic entropy production of the brain, even at rest. A promising future direction could involve quantifying entropy production across various task conditions as a measure of cognitive load [66]. This would involve investigating changes during tasks and determining if the control cost can explain the variation in energy the brain needs to supply for differing cognitive demands. Several other metrics have already been proposed to quantify the cognitive loads from microstate analysis, such as the entropy rate or the Hurst exponents [67,68]. However, these measures lack a clear interpretation regarding transition costs, as they do not account for dynamics. Our method, in contrast, estimates not the cognitive load during a single condition but the cost of transitioning between different conditions.

It would be further interesting to explore whether pathological conditions could influence the control cost. For instance, in the case of stroke, there have been documented changes in the microstates [69,70] and, more generally, in the dynamics of metastable states [25]. Additionally, the functional repertoire of macroscopic brain dynamics has been found to be reduced in Amyotrophic Lateral Sclerosis [71] and Parkinson's disease [72]. Therefore, it is intriguing to consider whether this altered flexibility is associated with an increase in control costs, which our framework could estimate. Furthermore, different conditions may affect distinct regions of the brain, resulting in alterations across various domains [73]. Consequently, it is reasonable to assume that the cognitive cost will be particularly higher for tasks impaired

due to specific neural alterations. Therefore, investigating individual differences in microstate transition cost in different groups (e.g., strokes), or applying it to tasks where the evaluation of cognitive demands is not known, are all interesting avenues to pursue in future research.

## Materials and methods

### Dataset

We re-analyzed the continuous EEG data collected in a recent study from our lab [74]. In that study, we aimed to investigate the neural correlates of cognitive control in resolving the interference between competing responses. To this aim, EEG signals were recorded from 44 participants during a 4-min resting state session and while they performed a spatial Stroop task requiring mouse responses and comprising blocks with three PC values (25%, 50%, and 75%) to manipulate different levels of cognitive control engagement (Fig 1A; see[74] for details about the task and procedure). Briefly, the data were recorded at 500 Hz with 64 electrodes mounted according to the 10–10 system. All electrodes were references to FCz during the recording. A standard ICA-based preprocessing was performed to correct for eye movements, blinks, and muscular activity based on scalp topography, dipole location, evoked time course, and the power spectrum of the components [74].

### EEG microstate-based analysis

Preprocessed EEG data were further bandpass filtered (1–40 Hz) and downsampled at 250 Hz.

Microstate analysis was performed using the open-source package "Pycrostates" [75]. Microstate analyses followed the modified k-means clustering algorithm [76,77]. First, a subject-level analysis was performed by extracting local maximal values (peaks) of the global field power (GFP) from each EEG recording. GFP was calculated as the standard deviation of the amplitude across all channels at each time point. EEG maps at GFP peaks are reliable representations of the topographic maps because of their high signal-to-noise ratio [78].

For each participant, we randomly extracted the same number (10000) of GFP peaks, that were subjected to clustering. Then, the individual topographies are used in the group-level analysis to fit a second clustering algorithm. The optimal number of clusters ($K^* = 7$) was determined using the cross-validation criterion, which minimizes the variance of the residual noise (see S1 Text). The centroids of the $K^*$ clusters identify the group-specific microstate templates (Fig 1B). Both the individual and group maps are reported in the online repository.

The group templates were then fitted back to the preprocessed EEG recordings. The EEG map at each time point was labeled according to the map with minimum Euclidean distance, equivalent to the highest absolute spatial correlation. Thereafter, EEG maps were converted into microstate sequences ($k_t$). For each EEG recording, we characterized the probability distribution of the microstates ($\pi$) under each condition. As the order of presentation of congruent and incongruent trials within each block was randomized, the microstate distribution for the task conditions was calculated from the pooled trials, excluding inter-trial intervals and boundary microstates. In contrast, the resting distribution was estimated during the initial resting phase. To ensure consistency, the resting phase was also divided into 2-second windows (i.e., the duration of a single trial). In addition, we compute the joint probability distribution $Q_{ij}$ for two consecutive steps $i$ and $j$ during the resting period (i.e., $Q_{ij} = Prob[k_{t-1} = i; \; k_t = j]$).

### Brain transition cost

To quantify the cost of transitioning from resting to task, we applied the Schrödinger bridge problem [36,39,60,79] (Fig 1C). We assumed that the brain dynamic unfolds over time along

trajectories of discrete states (i.e., the microstates), namely $K_0^T = (k_0, k_1, \ldots, k_T)$. Similarly to [39], we denoted by $q(K_0^T) = q(k_0, k_1, \ldots, k_T)$ the probability distribution of the trajectories at resting. The resting condition is also characterized by the (marginal) probability distribution of microstates $\pi^0$. To accommodate task demands, the brain has to modulate the probability distribution of the microstates, which we defined as $\pi^T$ for the target (task) probability distribution. The trajectory linking the resting to the task condition can be similarly characterized by the probability distribution $p(K_0^T)$. The control cost can thus be defined as the Kullback-Leibler (KL) divergence between the modulated $p(K_0^T)$ and spontaneous $q(K_0^T)$ trajectory distributions. This transition cost reflects how different the modulated trajectories are from the resting ones. Since estimating the probability distributions over the trajectories is infeasible, we can still infer the most probable trajectory as the Schrödinger bridge [39]. The Schrödinger bridge problem finds the most likely path linking the initial and target distribution given the prior stochastic evolution of the system by minimizing the Kullback-Leibler divergence between the two distributions.

In mathematical terms, the transition cost can be thus computed as

$$Cost = min_{P_{ij} \in U} D_{KL}(P_{ij}, Q_{ij}) = min_{P_{ij} \in U} - \sum_{ij} P_{ij} log Q_{ij} + \sum_{ij} P_{ij} log P_{ij} = min_{P_{ij} \in U} \sum_{ij} C_{ij} P_{ij} - H(P_{ij}),$$

where $Q_{ij}$ is the joint probability distribution for two consecutive timesteps at rest, $H(P_{ij}) = -\sum_{ij} P_{ij} log P_{ij}$ is the information entropy, and the minimization is constrained over the matrix spaces $U = \{P_{ij} > 0 \& \sum_{ij} P_{ij} = 1 | \sum_j P_{ij} = \pi_i^0 \& \sum_i P_{ij} = \pi_j^T\}$ (see [39,60] for the detailed mathematical derivation).

Interestingly, $C_{ij} = -log\, Q_{ij}$ plays the role of a transportation cost matrix, while each $P_{ij}$ that satisfies the constraints in U corresponds to a feasible trajectory linking the resting to the task condition. Indeed, the Schrödinger bridge problem can be recast as an (entropy-regularized) optimal transport problem [60]. Intuitively, to supply the needed cognitive demand, the brain has to modulate its dynamics, which results in a modulation of microstate distribution. In other words, the distribution of some microstates would be enhanced, while others would be suppressed. Thus, the brain has to "transport" some mass (i.e., microstate probability distribution) into another. How much mass is moved from each supply (i.e., resting) location to each demand (i.e., task) location is defined as the "transportation plan" and is encoded in the matrix $P_{ij}$. The transportation cost matrix C then represents the cost of transporting one unit of mass along each supply-demand pair. Solving the optimal transport problem means finding the transportation plan $P_{ij}$ that minimizes the total cost (with an entropic regularization term) while satisfying constraints like the given initial, supply, and final, demand distributions and non-negativity constraints (i.e., ensuring that negative values are not allowed in the transportation plan).

This is a strongly convex optimization problem, therefore the existence and uniqueness of the optimal solution are guaranteed. Such an optimal solution can be iteratively determined in an efficient way using the Sinkhorn algorithm [80].

## Statistical analysis

The statistical analyses were performed in Matlab using ad-hoc scripts.

First, to identify the microstates that were specifically more (or less) expressed at rest, we performed for each microstate a two-tailed one-sample t-test against .1429 (i.e., the chance probability reflecting a uniform distribution across the seven microstates, corresponding to 1/7) on the participants' probabilities of observing each microstate at rest.

Next, to investigate the task-dependent reconfiguration of microstate probability, we performed a linear mixed effects model for each microstate, including in the fixed part of the model the predictors for the stimulus congruency (incongruent vs, congruent trials, coded as

-.5 and .5, respectively), the PC level (PC25, PC50, and PC75, coded as -.5, 0, and .5, respectively), and their interaction. Consequently, the intercept effect denoted an overall modulation from the resting state to task execution. It is important here to note that we expected the PC level to have a linear, graded impact on the analyzed measures, in line with recent findings from our lab[46,47]. The same predictors were also included in the random part of the model along with the participants' random intercepts (i.e., we used a full random model). As a dependent variable, we computed the change in the probability distributions in the six different conditions (derived from the combination of the Congruency and PC levels) with respect to the resting state. We report the estimated coefficient ($b$), as well as the corresponding standard errors ($SE$) and $t$ and $p$ values for each fixed effect. We calculated the $p$ values by using Satterthwaite's approximation of degrees of freedom, which was also used to compute the corresponding effect size estimates (expressed as Cohen's $d$). The obtained results were then corrected for multiple tests ($n = 28$) by using the false discovery rate correction. After having fitted such a model, we performed a residual analysis, which verified the assumptions of homoscedasticity and normality of the residuals and did not reveal relevant signs of stress in model fitting.

The same analytical approach was used to investigate whether transition costs were modulated by task demands, by performing a similar full-random linear mixed effects model on the participants' transition costs, with the congruency by PC interaction in both the fixed and random part. Moreover, in order to investigate whether transition costs were related to task performance, we first computed the difference between transition costs in incongruent and congruent trials, corresponding to the Stroop effect in transition costs, and analyzed them with a linear mixed effects model with participants' Stroop effects in response times as a continuous predictor, which was also included in the random part of the model as participants' random slopes, along with the participants' random intercepts. It is important here to note that the Stroop effect is the standard measure of the interference resolution ability in the cognitive control literature.

## Supporting information

**S1 Fig. Selection of the best number of microstates using the cross-validation criterion.**
(*left*) Fraction of total variance (GEV) explained by the microstates. (*center*) Residual noise.
(*right*) Cross-validation (CV) as a function of the number of microstates (N states).
(PDF)

**S2 Fig. Behavioural results of the spatial Stroop task.** Distribution of response times (RT) for the 44 participants during each task condition.
(PDF)

**S3 Fig. Global modulation of microstate distributions during the spatial Stroop task.** Distribution of Kullback-Leibler divergence ($D_{KL}$) between the task ($\pi_{task}$) and resting ($\pi_{rest}$) for the 44 participants.
(PDF)

**S1 Table. LMM results for microstate occurrences.** For each microstate, a linear mixed model was implemented to test whether the change in the probability distribution during task execution compared to the resting state is modulated by the stimulus congruency, the PC level, and their interaction.
(PDF)

**S1 Text. Modified k-means clustering.** Mathematical details on the modified k-means clustering algorithm.
(PDF)

## Author Contributions

**Conceptualization:** Giacomo Barzon, Ettore Ambrosini, Antonino Vallesi, Samir Suweis.

**Data curation:** Ettore Ambrosini, Antonino Vallesi.

**Formal analysis:** Giacomo Barzon, Samir Suweis.

**Investigation:** Giacomo Barzon, Samir Suweis.

**Methodology:** Giacomo Barzon, Samir Suweis.

**Project administration:** Antonino Vallesi, Samir Suweis.

**Software:** Giacomo Barzon.

**Supervision:** Ettore Ambrosini, Antonino Vallesi, Samir Suweis.

**Visualization:** Giacomo Barzon.

**Writing – original draft:** Giacomo Barzon, Samir Suweis.

**Writing – review & editing:** Giacomo Barzon, Ettore Ambrosini, Antonino Vallesi, Samir Suweis.

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
