## [Decision Letter · Decision Letter 0]

22 Feb 2024

Dear Mr Barzon,

Thank you very much for submitting your manuscript "EEG microstate transition cost correlates with task demands" for consideration at PLOS Computational Biology.

As with all papers reviewed by the journal, your manuscript was reviewed by members of the editorial board and by several independent reviewers. In light of the reviews (below this email), we would like to invite the resubmission of a significantly-revised version that takes into account the reviewers' comments.

Three expert reviewers have provided thorough reviews on your manuscript. While they are all interested in your study, more serious concerns are voiced regarding methods and analyses, but also for the interpretation of findings. In sum, a strong revision will be needed to address these concerns adequately.

We cannot make any decision about publication until we have seen the revised manuscript and your response to the reviewers' comments. Your revised manuscript is also likely to be sent to reviewers for further evaluation.

Sincerely,

Christoph Strauch

Academic Editor

PLOS Computational Biology

Lyle Graham

Section Editor

PLOS Computational Biology

Three expert reviewers have provided thorough reviews on your manuscript. While they are all interested in your study, more serious concerns are voiced regarding methods and analyses, but also for the interpretation of findings. In sum, a strong revision will be needed to address these concerns adequately.

Reviewer's Responses to Questions

**Comments to the Authors:**

Reviewer #1: Barzon et al. describe a new approach to develop a quantitative measure of "cognitive effort" using EEG microstates. The method is tested on a previously published dataset in which healthy participants perform a spatial Stroop task at three different levels of cognitive control demand.

They find:

- different microstate occurrences for different tasks

- transport cost depends on stimulus type, cognitive control level, and their interaction

- transport cost differences (Delta cost) correlates positively with reaction time (RT)

Strengths:

- The authors address a relevant question, i.e. how to estimate cognitive effort from neurophysiological data.

- They present a novel and innovative method. Transport theory has not been applied to EEG microstate data.

- The method is well described and reproducible.

- Microstate methodology follows standard procedures including an estimation of the optimum number of clusters (CV criterion).

- The experimental design has a tunable parameter PC (proportion of congruency) that gives a clear hypothesis about the actual cognitive demand.

- The results are well documented, quantified and statistically evaluated.

- The dataset has a good sample size, n=44.

While the overall approach and the presented results promise a valuable and insightful publication, there is a number of open issues, listed below.

- Existing strategies to estimate cognitive load should be discussed in more detail. Previously used methods include spectral analyses but also EEG microstate analyses, in particular Jia et al., https://doi.org/10.1038/s41598-021-03577-1. To validate their new method, the authors should compare their results with at least one previously published approach (e.g. entropy rate, Hurst exponent).

- The proposed transition cost measure has an interesting interpretation in terms of brain state switches. Whether these transition dynamics (Q_ij) are really relevant could be tested by comparing with a simpler approach that looks at the shape

of the microstate distribution only. For example, does the entropy of the microstate distribution alone (in each condition) perform worse than the proposed measure that takes into account Q_ij?

- p.4, Fig. 1: "found in the literature [Michel et al. 2018], and we labeled them accordingly (from A to E)". Microstate class E (ms-E) shown in Fig. 1b is very different from ms-E in Michel et al., NeuroImage, 2018, or in Custo et al., 2017. The authors should discuss this and consider using a different label for this map.

- p.8, microstate backfitting: "The EEG map at each time point was labeled according to the map...". Does this imply that maps were back-fitted at each time point without any temporal smoothing (interpolation, parametric smoothing, minimum duration etc.)? If yes, this should be made explicit as smoothing is very common in the microstate community.

- The authors use the term "occurrence" throughout the manuscript. While it becomes clear what they are referring to, I would suggest to switch to "coverage" or "distribution" as occurrence is a commonly used but different microstate parameter that refers to the frequency with which a given microstate class occurs and therefore has units of frequency (1/s). This could easily confuse readers used to the 'standard' terminology.

- The authors identify ms-A as indicative of the resting condition (e.g. Fig. 2). This is very unusual, ms-A is commonly associated with task performance and rest is often associated with ms-C (ms-D), probably representing posterior dominant

resting state rhythms (alpha). This must be discussed and further assessed. Is this attributable to the relatively large number of clusters (K=9)? Does the finding persist in the case K=4?

- Please provide more quantitative information about the analysed trials. Apparently, microstate distributions were obtained for each presented stimulus. How long were these trials on average and how many microstates were found during one stimulus/response trial? From Tafuro et al., I understand that participants had 750 ms to initiate movement, 2000 ms to complete, and 1500 ms blank post-stimulus. Let's say an average trial lasts about 2000 ms (assuming that participants

complete in much less than the given 2000 ms) and the average microstate duration is 100 ms. This gives only 20 microstates per trial and as little as 2 samples/histogram bin for K=9 microstates. Please provide the actual number of samples that went into the estimated microstate distributions. Within a block (high, medium, low), were the microstate distributions for all

C- (or I-) trials pooled? Were microstates at trial boundaries excluded?

- p.4: "In essence, this cost was calculated as the disparity between the spontaneous microstate dynamics during the resting phase and the bridge".

This is where my most serious doubt lies. While the approach is clear and meaningful in general, this does not seem to reflect the experimental reality of the dataset. From Fig. 1a, I understand that the resting phase occupied the first 4 minutes,

followed by the task phase that switches between high/medium/low blocks and C/I stimuli within each block. This means that the actual brain state transitions occurred between different task conditions (e.g. C-M and I-M, or between high and medium), but not between rest and task.

Is the bridge calculation rest-task meaningful in this context?

Does the approach still work when you calculate transition costs between the different tasks as they occurred during the actual experiment?

Do the authors assume that subjects return to the resting state between consecutive stimuli? If yes, can this be justified from the literature?

- p.5 "our results hint at a macroscopic entropy production..." - consider moving this to the Discussion section.

- "stimulus type" is only used twice on page 6. As it is an important part of the results, please introduce the term in the Methods section.

- Fig. 4a: The bracket on the right (single asterisk) seems to compare C-low and I-low. Is this a mistake and should the bracket indicate the comparison I-high and I-low instead? Please clarify.

- Fig. 4b: The correlation is not very clear from the plot. "revealed a significant positive correlation within this distribution" -

does this refer to the whole data cloud shown in Fig. 4B? Can the authors please add a line indicating the correlation?

- The Discussion section is extremely short regarding the neurobiological interpretation and the potential practical value of the results.

Points that should be addressed include:

a) Why does ms-A appear as an indicator of the resting state (and not ms-C, as reported in the literature)?

b) Fig. 4a, why are transition cost differences more pronounced for the easier stimulus type C (congruent)? Shouldn't the transition cost differences be larger for the more demanding incongruent (I) type?

c) Fig. S4 shows that RT is large for I-low and C-high. Doesn't this mean that RT is an inconsistent measure of cognitive demand?

d) Given the variance, is the effect size observed for incongruent stimuli shown in Fig. 4a relevant for practical use at all? There seems to be (almost) no change in transition cost although the PC varies between 75% and 25%.

e) Although statistically significant, is the correlation shown in Fig. 4b of any practical value? How large is the correlation coefficient? I hypothesize that the correlation is significant only due to the congruent trials. The cost for incongruent tasks looks constant in Fig. 4a, so the difference congruent-incongruent follows the congruent data. Why did the authors assess the difference values, are they of any biological relevance (are they used in the literature?).

Overall, as the authors promise "fresh perspectives for physiologically describing cognitive effort", the results require a more critical evaluation. There are some positive correlations for congruent trials but the discrepancy between congruent and incongruent tasks makes the results much less convincing (unless explained further).

Minor points:

---------------

- The greek letter pi occurs as both upper and lower case letters throughout the manuscript and Figures, please use one form only.

- The term in Fig. 1c is "min D_KL", on p. 9 it's "min KL"

- Fig. S2: See above, "occurrence" can be confusing as most microstate papers define occurrence differently and a unit of 1/s is expected.

Reviewer #2: This manuscript tries to link the construct of cognitive effort to the transition cost between multiple brain states. This research question is answered by using electrophysiological indexes for those quasi-stable brain states, i.e. micro-states, in the context of a cognitive control demanding task, a spatial Stroop task. The authors use the difference in microstates occurrence between a resting-state and the three different cognitive control level conditions of the task to show evidence for the link between cognitive effort, task conditions and ultimately behavior. In this paper, the proxy used for cognitive control is the transition cost between the different states that is estimated through a novel method for EEG.

At the general level, the paper presents an interesting approach, namely, how to assess cognitive effort through the estimation of transition costs among quasi-stable brain states. The paper, however, presents a strong lack of details in the methods but also the reported results. This lack impacts not only the replicability of the method to other contexts but also simply the understanding of most analysis in the paper. I come back to this issue among the several major points that I think need to be addressed. Parallel to this lack of methodological aspects, or maybe related, it looks like the statistical analysis chosen by the authors doesn't allow to tell whether the data supports the conclusion of the authors.

Because of these problems my main comments on this manuscript are on the methodological side rather than on the research questions per se. In the following, I detail my main concerns for each section of the results.

On the microstates occurrences: 

- I am quite surprised by the lack of variance among the state occurrences. Based on Fig S2 (which would better sit in the main text given its importance) it does look like every state has a probability of roughly 1/9 (and 9 happens to be the number of microstates). This uniform distribution of state occurrences doesn't seem to be true in other microstates paper (e.g. Milz et al. 2016) and, to me, questions whether the microstates extracted are really task related. I would suggest that the authors explore ways of assessing whether their clustering is robust, e.g. resampling in the context of microstates https://pycrostates.readthedocs.io/en/latest/generated/auto_tutorials/cluster/10_subject_level_with_resampling.html

- On the linear mixed model. First I am surprised that the author chose a linear model and not a generalized linear model as the normality assumption of the residuals implied by a linear model cannot be respected in a change of probability of occurrence. That is, maybe this assumption is reasonable if those changes in occurrences are close to 0.5 but this is clearly not the case (Fig. S2). More importantly, the model description (also in the supplementary material) is insufficient to understand and evaluate the statistical model. What contrast coding was applied to the factors (e.g. reference/intercept is compatible condition? how were the three categories of control level coded?), what random structure was chosen?  Moreover the post-hoc test applied could be avoided by having a better modelization. In this case, the Bonferroni correction will increase type II errors which might hide interesting results from this analysis.

Transportation cost matrix: I fail to understand the method. This might be due to my mathematical limitations and/or to an under-reporting of the method. In either case, this method being the core of the paper, it should be explained more thoroughly (e.g. what is P_{ij} in the equation on p. 9) and clearly (e.g. while I think I understand the parenthesis for Q_{ij}, I fail to see the link with the associated sentence).

transition cost reflects task demand:

- I don't understand why the authors choose a two-way ANOVA. It seems to me that the measures in this case are also repeated across participants. Hence at the minimum a repeated measure ANOVA needs to be conducted or, for more coherence with the microstate occurrence analysis, a linear mixed model. Without this control it is impossible to support this analysis, even more so in the absence of any index usually reported along with the p-values (i.e. no F statistic or degrees of freedom). 

- For the link between reaction time and transition cost also the analysis is hard to understand because of the lack of reporting. The authors choose to compute correlations for each participant between RTs difference and cost. Is this between control levels? If so, I don't think a correlation is appropriate as 1) it is only for three points, 2) no guarantee exists on the linear link between RTs and cost difference across the three levels and 3) the second step analysis of the t-test ignores the uncertainty in the correlation. Assuming that point 2) is verified a better modelling approach would be to use again a linear mixed model and report the coefficient along the p-values. This would then resolve point 3) and, based on my understanding of what the authors did, be a proper test for the hypothesis laid out in this paragraph.

Minor:

- p.8 "To this aim, we recorded EEG in 44", implies that the data was recorded for this study

- p.5 last sentence of first paragraph, "Instead", poor wording?

- The terms High, medium and low to describe 25, 50 and 75 % of congruent trials is a conflict in itself. It would be easier to read if the proportion of incongruent trials was reported instead

- In p.2 I had difficulties following what exactly transition cost were defined as in relation to the research question, the Result section is however clearer. For the sake of readability, the authors may want to rewrite this section by explicitly relating to microstates in resting vs task.

- p. 6 last paragraph,  "subjective performance" to describe RT seems surprising as RT doesn't usually qualify as a subjective measure.

- p. 8 Dataset section insufficiently describes the dataset, how many electrodes, what reference was used, the manuscript should be self-standing for such aspects

Sincerely,

Gabriel Weindel

New references:

Milz, P., Faber, P. L., Lehmann, D., Koenig, T., Kochi, K., & Pascual-Marqui, R. D. (2016). The functional significance of EEG microstates—Associations with modalities of thinking. _Neuroimage_, _125_, 643-656.

Reviewer #3: This research paper focuses on the dynamics of the brain, describe as microstates as observed in EEG, and their role with respect to cognitive processes. In particular, the authors investigate how brain activity correlates with the level of cognitive exertion during tasks by employing EEG data. These studies apply the principles of optimal transport theory and the Schrödinger bridge problem to examine the changes and dynamics in the brain as it responds to varying cognitive challenges.

I think this paper is sound, and I do not have strong objections to it. In particular, I thought it was interesting to note that employing optimal transport theory aids in comprehending the brain's dynamics and the cognitive effort required during tasks by transforming the Schrödinger bridge problem into an entropy-regularised optimal transport issue. This method facilitates the adjustment of brain dynamics to accommodate cognitive needs by altering the distribution of EEG microstates, intensifying some while diminishing others. By identifying the most cost-efficient transportation plan, which incorporates an entropic regularisation component, researchers can study the variations in EEG microstates throughout tasks, identifying greater costs linked to cognitive shifts. The transportation cost matrix represents the cost of moving mass between supply (resting) and demand (task) locations, offering insights into the brain's strategies for handling and adapting to different cognitive demands. This approach presents an innovative method for evaluating cognitive effort through brain data, underlining the connection between patterns of neural activity, cognitive strain, and behavioural outcomes.

At the end of the article, the whole scenario is clear, but at the level of the following sentence, I was confused. Hence, in my opinion, it would help to explain more clearly why it is possible to not use a mathematical modelling.

<<instead, advantage="" approach="" cost="" directly="" estimating="" from="" neurophysiological="" of="" offers="" our="" reconfiguration="" recordings="" the="">>

It would be interesting to also have the same analysis in a pathological context to explore whether pathological conditions could influence the control cost, because has been observed that in pathological conditions we have a reduced flexibility in brain dynamics linked to different sequences of patterns.

For future research, can be a good idea to focus more on brain dynamics and study the evolution of these microstates across EEG data, implementing a mathematical model which, starting from a realistic framework activity, can also implement the Schrödinger bridge problem to reproduce a similar analysis. In particular, some papers share of the motivations with these paper, and have explored complexity and flexibilithy in pathology, such as:

Cipriano, Lorenzo, et al. "Flexibility of brain dynamics is increased and predicts clinical impairment in Relapsing-Remitting but not in Secondary Progressive Multiple Sclerosis." medRxiv (2023): 2023-07; Polverino, Arianna, et al. "Flexibility of fast brain dynamics and disease severity in amyotrophic lateral sclerosis." Neurology 99.21 (2022): e2395-e2405; and Sorrentino, Pierpaolo, et al. "Flexible brain dynamics underpins complex behaviours as observed in Parkinson’s disease." Scientific reports 11.1 (2021): 4051. In my opinion, this is relevant literature in this context.

In the abstract, I would suggest describing in a very concise way what is the Stroop task to give immediately the main idea of the work.

In the introduction, after the following sentence, you can add a reference:

<< At the macroscale, brain activity is characterized by spatially distributed groups of regions that exhibit temporally correlated activity and co-activate during behavioral tasks, thus acting as functional networks>>.

The results and methods used are well described, clear and well explained. The statistical analysis done is sufficient.

All in all, I find this is a sound piece of scientific work, and I congratulate the authors.</instead,>

**Have the authors made all data and (if applicable) computational code underlying the findings in their manuscript fully available?**

Reviewer #1: Yes

Reviewer #2: Yes

Reviewer #3: Yes

PLOS authors have the option to publish the peer review history of their article (what does this mean?). If published, this will include your full peer review and any attached files.

Reviewer #1: No

Reviewer #2: **Yes: **Gabriel Weindel

Reviewer #3: No
---

## [Decision Letter · Decision Letter 1]

5 Sep 2024

Dear Mr Barzon,

Thank you very much for submitting your manuscript "EEG microstate transition cost correlates with task demands" for consideration at PLOS Computational Biology. As with all papers reviewed by the journal, your manuscript was reviewed by members of the editorial board and by several independent reviewers. The reviewers appreciated the attention to an important topic. Based on the reviews, we are likely to accept this manuscript for publication, providing that you modify the manuscript according to the review recommendations.

The reviewers and myself have found the manuscript to have much improved in the revision. There are a few comments left to address, but as these are fairly minor I don't think we have to bother the reviewers again, provided that these points are addressed properly in a minor revision. Congratulations on this very interesting manuscript!

Sincerely,

Christoph Strauch

Academic Editor

PLOS Computational Biology

Lyle Graham

Section Editor

PLOS Computational Biology

The reviewers and myself have found the manuscript to have much improved in the revision. There are a few comments left to address, but as these are fairly minor I don't think we have to bother the reviewers again, provided that these points are addressed properly in a minor revision. Congratulations on this very interesting manuscript!

Reviewer's Responses to Questions

**Comments to the Authors:**

Reviewer #1: The revised version of Barzon et al. has answered most of my questions.

I still have a few comments and minor requests.

1) Re entropy rate:

I thank the authors for their additional analyses. The finding of a maximum entropy rate during rest is quite surprising, despite the literature cited by the authors. None of the three cited papers computes the entropy rate of a neurophysiological time series (EEG or BOLD). Escrichs et al. do not use entropy at all.

The entropy rate of a microstate sequence is expected to be lower in the resting state as the EEG becomes faster during cognition, as reported in Jia et al.

Together with the unusual finding of microstate-A dominance in the resting state, I wonder if the resting state was an eyes-open resting state or had low alpha power for some other reason ? I couldn't find information about eyes open/closed. Please add this.

I think the reader should be informed that the resting state in this study has these somewhat unusual features, to facilitate future comparisons. Please add

this to the Discussion.

2) Re KL-divergence:

The additional findings presented in Fig. S3 show that KL-divergence between the microstate distributions alone seems to perform quite well as an estimator of cognitive load. Re-iterating my request from the first review, could the authors please quantify which method is more effective in measuring cognitive cost? E.g. by comparing the effect size. I think this should also be added to the Discussion where Fig. S3 is just briefly mentioned. The reader should learn that it performs with a similar efficacy (or better than?) the more advanced Schroedinger bridge method although KL-divergence ignores the transition path. The simple method performs nicely and is closely related to the authors' approach. It shows that cognitive load is already encoded in the distance between the distributions, independent of the optimum path found between them.

3) Re entropy production:

Not sure if the author comment "However, to our knowledge, this measure has not yet been explored in the EEG field" was meant to say in the EEG microstate field?

Anyway, the technique has been used for non-human primate EEG data (https://doi.org/10.1093/cercor/bhac177) and has also been used to demonstrate

irreversibility and non-equilibrium dynamics in EEG microstate time series in Hermann et al. (https://doi.org/10.1007/s10548-023-01023-1).

These should be cited in the Discussion.

Minor:

The time axis in Fig R4 cannot be ms but thank you very much for clarifying the question.

Thanks for all the extra work that went into the revision. This is a very interesting approach and I'm looking forward to reading future research with it.

Reviewer #2: Congratulations to the authors for the improvement of the manuscript. I am now much more convinced by the results and the readability of the paper has greatly improved. I only have a few minor points left:

- The new sentence in the abstract could be merged with the previous one for something more concise but still informative on the nature of the task

- The results of the mixed models in section "Microstate reconfiguration during task" are now hard to follow, maybe the authors could only report coefficients and SE and report p values and cohen's d in a table). This would make these results clearer both for those who just want a quick summary and those who want to dig into the details and can look up the table.

- Bonini, Francesca, et al. "Action monitoring and medial frontal cortex: leading role of supplementary motor area." Science 343.6173 (2014): 888-891. Might be an interesting reference for the discussion for the modification of states G and F (3rd paragraph in the discussion)

Reviewer #3: I thank the authors for addressing my comments. I have no further remarks.

**Have the authors made all data and (if applicable) computational code underlying the findings in their manuscript fully available?**

Reviewer #1: **No: **Raw data is not available, as far as I can see. Ok for me though.

Reviewer #2: None

Reviewer #3: Yes

PLOS authors have the option to publish the peer review history of their article (what does this mean?). If published, this will include your full peer review and any attached files.

Reviewer #1: No

Reviewer #2: **Yes: **Gabriel Weindel

Reviewer #3: No

Figure Files:

Data Requirements:

Reproducibility:

References:

---

## [Editor Report · Decision Letter 2]

28 Sep 2024

Dear Mr Barzon,

We are pleased to inform you that your manuscript 'EEG microstate transition cost correlates with task demands' has been provisionally accepted for publication in PLOS Computational Biology.

Best regards,

Lyle J. Graham

Section Editor

PLOS Computational Biology

---

## [Editor Report · Acceptance letter]

3 Oct 2024

PCOMPBIOL-D-23-02005R2 

EEG microstate transition cost correlates with task demands

Dear Dr Barzon,

I am pleased to inform you that your manuscript has been formally accepted for publication in PLOS Computational Biology. Your manuscript is now with our production department and you will be notified of the publication date in due course.

With kind regards,

Anita Estes
